# GPT-4 can pass the Korean National Licensing Examination for Korean Medicine Doctors

**Dongyeop Jang**[1], **Tae-Rim Yun**[1], **Choong-Yeol Lee**[1], **Young-Kyu Kwon**[2,3], **Chang-Eop Kim**[1,4] *

**1** Department of Physiology, College of Korean Medicine, Gachon University, Seongnam, Gyeonggi-do, Korea, **2** Division of Integrated Art Therapy, School of Korean Medicine, Pusan National University, Yangsan, Gyeongsangnam-do, Korea, **3** Division of Longevity and Biofunctional Medicine, School of Korean Medicine, Pusan National University, Yangsan, Gyeongsangnam-do, Korea, **4** Department of Neurobiology, Stanford University School of Medicine, Stanford, California, United States of America

* eopchang@gachon.ac.kr

**Data Availability Statement:** The benchmark dataset, Korean National Licensing Examination, is available from the Korea Health Personnel Licensing Examination Institute (https://www.kuksiwon.or.kr/).

## Abstract

Traditional Korean medicine (TKM) emphasizes individualized diagnosis and treatment. This uniqueness makes AI modeling difficult due to limited data and implicit processes. Large language models (LLMs) have demonstrated impressive medical inference, even without advanced training in medical texts. This study assessed the capabilities of GPT-4 in TKM, using the Korean National Licensing Examination for Korean Medicine Doctors (K-NLEKMD) as a benchmark. The K-NLEKMD, administered by a national organization, encompasses 12 major subjects in TKM. GPT-4 answered 340 questions from the 2022 K-NLEKMD. We optimized prompts with Chinese-term annotation, English translation for questions and instruction, exam-optimized instruction, and self-consistency. GPT-4 with optimized prompts achieved 66.18% accuracy, surpassing both the examination's average pass mark of 60% and the 40% minimum for each subject. The gradual introduction of language-related prompts and prompting techniques enhanced the accuracy from 51.82% to its maximum accuracy. GPT-4 showed low accuracy in subjects including public health & medicine-related law, internal medicine (2), and acupuncture medicine which are highly localized in Korea and TKM. The model's accuracy was lower for questions requiring TKM-specialized knowledge than those that did not. It exhibited higher accuracy in diagnosis-based and recall-based questions than in intervention-based questions. A significant positive correlation was observed between the consistency and accuracy of GPT-4's responses. This study unveils both the potential and challenges of applying LLMs to TKM. These findings underline the potential of LLMs like GPT-4 in culturally adapted medicine, especially TKM, for tasks such as clinical assistance, medical education, and research. But they also point towards the necessity for the development of methods to mitigate cultural bias inherent in large language models and validate their efficacy in real-world clinical settings.

**Funding:** This work was supported by the National Research Foundation of Korea (NRF) grant funded by the Korea government (MSIT) (No. 2020R1F1A1075145 to Y.-K. K. and 2022R1F1A1068841 to C.-E. K.). The funders had no role in study design, data collection and analysis, decision to publish, or preparation of the manuscript.

**Competing interests:** The authors have declared that no competing interests exist.

## 1. Introduction

Traditional Korean medicine (TKM) is the medicine that has been traditionally practiced in Korea based on ancient Chinese medicine and constitutes Korea's unique medical system along with Western ("conventional") medicine [1]. Traditional Asian medicine, including TKM, and traditional Chinese medicine, has been used complementarily with Western medicine [2] significantly improving patient outcomes [3–5]. TKM emphasizes the importance of individualized diagnosis and treatment, taking into account the patient's unique symptoms and constitution. The decision-making process of TKM clinicians is complex and often depends on their clinical experience and intuition [6].

In recent decades, several studies have aimed to develop AI that models the decision-making process of TKM clinicians using rule-based methods and machine learning-based methods. Rule-based methods involve creating a set of rules based on expert knowledge and applying these rules to choose appropriate diagnoses and interventions for patients. For example, in the 2000s, semantic web and ontologies for TKM [7] and traditional Chinese medicine (TCM) [8,9] were constructed to express the relationships between entities in medicine including symptoms, diseases, and interventions, and to model decision-making processes. However, one of the limitations of such approaches is that the entities of TKM are intertwined in a non-linear relationship with each other [10], and it is difficult to represent the knowledge and reasoning processes of TKM. Moreover, much of the decision-making process of TKM clinicians is implicit [6], therefore it is difficult to explicitly define a general decision-making process.

Machine learning-based methods, on the other hand, involve training algorithms on large datasets of patient data to identify patterns and make predictions. For example, a stacked auto-encoder to categorize a clinical case for hypertension based on symptoms [11] or a text classification model to perform syndrome differentiation from TCM in medical records [12] have been developed. In addition, the process of classifying constitutional conditions has been modeled with extremely randomized trees and key features for constitutional classification have been identified [13]. Since the development of deep learning-based natural language models represented by Transformer [14], models utilizing natural language data have been developed, such as a BERT-based model to support diagnosis based on medical records in natural language [15] and a BERT with text-convolutional neural network (Text-CNN) based model to classify medical cases [16]. However, the development of medical/clinical AI is restricted by privacy concerns and the need for high expertise in domain knowledge [17]. In particular, terminology and inference of TKM are disparate from general medicine, so it has been believed that TKM-specific training is necessary to build models that can understand and utilize medical/clinical textual records in TKM for decision-making [18].

GPT-3.5 and GPT-4 are large language models (LLMs) created by OpenAI that have been trained based on a massive amount of text data by self-supervised learning [19]. These models have been designed to mimic human conversationalists but have recently become widely recognized for their impressive ability for natural language processing across multiple specialized areas. GPT-3.5, popularized by ChatGPT (https://chat.openai.com/chat), is a fine-tuned model from GPT-3 by reinforcement learning from human feedback [20,21]. GPT-4 was released on March 14, 2023, and is the fourth GPT model created by OpenAI. GPT-4 demonstrates human-level performance on a variety of professional and academic benchmarks [22]. In medicine, GPT-3.5 and GPT-4 have shown impressive performance on the United States Medical Licensing Examination (USMLE) [22,23], supporting the argument that LLMs without advanced training for medical texts, including GPT-4, can have an impact on medical practice [24].

In this study, we aim to evaluate the potential to use LLMs as AI that can perform decision-making in TKM. To this end, we input questions from the Korean National Licensing

Examination for Korean Medicine Doctors (K-NLEKMD) into GPT-3.5 and GPT-4 and conducted a thorough analysis of its responses.

## 2. Methods

### 2.1. K-NLEKMD

In the Republic of Korea, a TKM doctor must pass the K-NLEKMD under the administration of the National Health Personnel Licensing Examination Board (https://www.kuksiwon.or.kr/) and be licensed by the Minister of Health and Welfare to obtain legal status and practice. This examination evaluates the competency of prospective doctors, encompassing questions that assess competencies both as a TKM specialist and a general medical practitioner. It is developed by faculty members from medical colleges or universities, all of whom have at least 3 years of teaching experience in the field and have completed a questionnaire development workshop. Candidates must correctly answer at least 60% of the total questions and at least 40% of questions for each subject to pass the national examination [25]. The examination comprehensively evaluates a candidate's ability to recall medical knowledge, interpret presented clinical data, and provide appropriate care for patients in each situation.

Each question has a closed-form structure with a single best choice out of five choices. For this evaluation, we utilized examination questions from the K-NLEKMD administered in January 2022. The questions and answers are accessible upon request to the National Health Personnel Licensing Examination Board. We have confirmed that all questions cannot be searched on the web, suggesting that the language models were not directly trained on the questions used in the evaluation.

### 2.2. Language models and inputs

In this study, GPT-3.5 [26] and GPT-4 [22] were used for benchmarks and implemented with the OpenAI API (https://openai.com/blog/openai-api, accessed in September 2023). For questions containing tables, the contents of the table were input as text, separated by spaces. For questions that included images, we excluded the images from the input and used only the accompanying text, since our version of the model couldn't process images at the time of the experiment. To ensure memory retention and in-context learning don't influence the results, we reset the chat session for every trial and question.

### 2.3. Prompt engineering

We applied prompt engineering techniques to maximize the performance of LLMs for solving the TKM examination. The details and examples of prompts used in this study are described in S1 Table. In brief, four kinds of techniques were applied in the prompts: annotating Chinese terms in TKM, translating the instruction and question into English, providing exam-optimized instructions, and utilizing self-consistency [27] in the prompt. We recognized that the language model might struggle with TKM terms usually written in Chinese characters but presented in Korean script only in our questions. To address this, we provided annotations in Chinese characters for these TKM terms (S2 Table). Also, considering the gap between the amount of English-written texts and the amount of Korean-written texts is huge in the training datasets [28], it is reasonable to infer that LLMs would perform better with English input than with other languages. Therefore, we had the models translate either the prompt, the questions, or both and then re-input the translated text for answer prediction. Exam-optimized instruction is designed to reason 'step by step' manner, similar to a 'chain of thought' (CoT) approach [29], and select just one choice from the presented five choices for each question. Self-

consistency is a method inspired by ensemble models that combine the predictions from multiple models to generate a final answer [27], and we deduced the final answers of the model by choosing the most frequent response in the independently processed seven trials.

## 2.4. Classification for questions

For the 12 subjects: internal medicine (1) (focused on general internal diseases), internal medicine (2) (covering Shanghanlun and Sasang medicine, foundational theories in traditional Korean medicine), acupuncture medicine, public health & medicine-related law, dermatology & surgery, neuropsychiatry, ophthalmology & otorhinolaryngology, gynecology, pediatrics, preventive medicine, physiology (specialized in traditional Korean medicine), and herbology; there are 80, 32, 48, 20, 16, 16, 16, 32, 24, 24, 16, and 16 questions included respectively, as shown in Table 1. Furthermore, each question was labeled based on several criteria: the necessity of TKM-specialized knowledge for answering (204 questions), the presence of a table (13 questions) or figure (46 questions), the subject from which the question originated, and the competency type it aimed to measure (114 questions for recall, 99 for diagnosis, and 127 for intervention). TKM-specialized knowledge is defined as knowledge that is intrinsic to TKM, distinguishing it from what other medical practitioners might know. Specifically, this examination evaluates candidates' proficiency as TKM specialists, alongside their general healthcare provision capabilities. To clarify, we anticipated that questions would require TKM specialized knowledge if they encompassed TKM etiological or staging concepts (like qi, yin, heat deficiency), treatment-related concepts (such as herbal medicine, acupuncture points), and specific TKM prescriptions (for instance, spleen-deficiency, so-gal, which parallels diabetes and its complications in Western medicine). Diagnosis-based questions included those that asked for a diagnosis for each case, or what additional information was needed to diagnose. Intervention-based questions include questions about the most appropriate pharmacological or acupuncture treatment for each case or questions about the most appropriate prevention methods. (Table 2). The questions were classified by five certificated TKM doctors, and three of them have more than five years of experience in teaching TKM in the college of TKM.

**Table 1. Subjects in the Korean National Licensing Examination for Korean Medicine Doctor.**

| Subject | The number of questions |
|---|---|
| Internal medicine (1) * | 80 |
| Internal medicine (2) ** | 32 |
| Acupuncture medicine | 48 |
| Public health & medicine-related law *** | 20 |
| Dermatology & Surgery | 16 |
| Neuropsychiatry | 16 |
| Ophthalmology & Otorhinolaryngology | 16 |
| Gynecology | 32 |
| Pediatrics | 24 |
| Preventive medicine | 24 |
| Physiology **** | 16 |
| Herbology | 16 |
| Total | 340 |

\* A subject on the specialty dealing with general internal diseases

\** A subject on the specialty of Shanghanlun and Sasang medicine, classical theories in traditional Korean medicine

\*** Limited to public health and medicine-related laws enforced in the Republic of Korea

\**** Physiology specialized in traditional Korean medicine

**Table 2. Classification for questions in the Korean National Licensing Examination for Korean Medicine Doctor.**

| | | TKM-specialized knowledge | | Sum |
|---|---|---|---|---|
| | | Yes | No | |
| **Competency type** | Recall | 59 | 55 | 114 |
| | Diagnosis | 23 | 76 | 99 |
| | Intervention | 122 | 5 | 127 |
| **Table** | Yes | 5 | 8 | 13 |
| | No | 199 | 128 | 327 |
| **Image** | Yes | 18 | 28 | 46 |
| | No | 186 | 108 | 294 |

## 2.5. Encoding of answers

Each question has five choices and only one of them is the correct answer. If the response of the language model is not the same as the correct answer, it is counted as the wrong response. If the model output the number of the correct answer choice or if it did not mention a number but the response corresponded to the correct answer choice, it was accepted as correct. On the other hand, if the model provided a reasonable explanation for each choice but did not confirm the answer, the answer was not accepted as correct. If the model responded that there was more than one answer or that there was no correct answer, we treated the response as incorrect. If the model refused to provide an answer because it was not authorized for medical diagnosis or prescription (e.g., "I am an artificial intelligence model and do not have specialized medical knowledge or diagnostic abilities. You should consult a medical professional for a diagnosis."), or because answering a test question would violate academic integrity (e.g., "It is a violation of academic ethics to use a language model to solve test questions."), the response was discarded and the question was resubmitted in a new session. When assessing the model's average capabilities, we typically had it tackle the same problem five times independently. We then calculated and reported the mean score and its standard deviation over these repetitions. To evaluate the performance based on self-consistency, we used the most frequent answer out of the seven repetitions for each question as the model's main response. Then, we graded it based on that.

## 3. Results

### 3.1. Overall performance on the K-NLEKMD

To measure the overall performance of language models on TKM, we measured the accuracy for entire questions from the KNLEKMD. As a result, GPT-4 with the most optimized prompt achieved an accuracy of 66.18%, indicating that the accuracy is over the pass mark of the examination of 60%. Comparing the accuracy of GPT-3.5 and GPT-4, the accuracy of GPT-4 is higher compared to GPT-3.5 (S1 Fig). Given the superior performance of GPT-4 over GPT-3.5, all subsequent analyses and results presented in this paper will focus on GPT-4.

We observed that the model's accuracy consistently improved as we introduced various techniques. Without applying any prompt techniques, the model achieved an accuracy of 51.82%. Meanwhile, annotating Chinese terms, translating the instruction and question into English, providing exam-optimized instructions, and utilizing self-consistency improved accuracy by 57.59%, 60.47%, 63.65%, and 66.18%, respectively (Fig 1). This indicates that each of these modifications plays a role in enhancing the model's accuracy. In the examination for the same questions for candidates for the TKM doctors, the average total accuracy was 76.7%, suggesting that the performance of GPT-4 on TKM is not yet at the level of human experts.

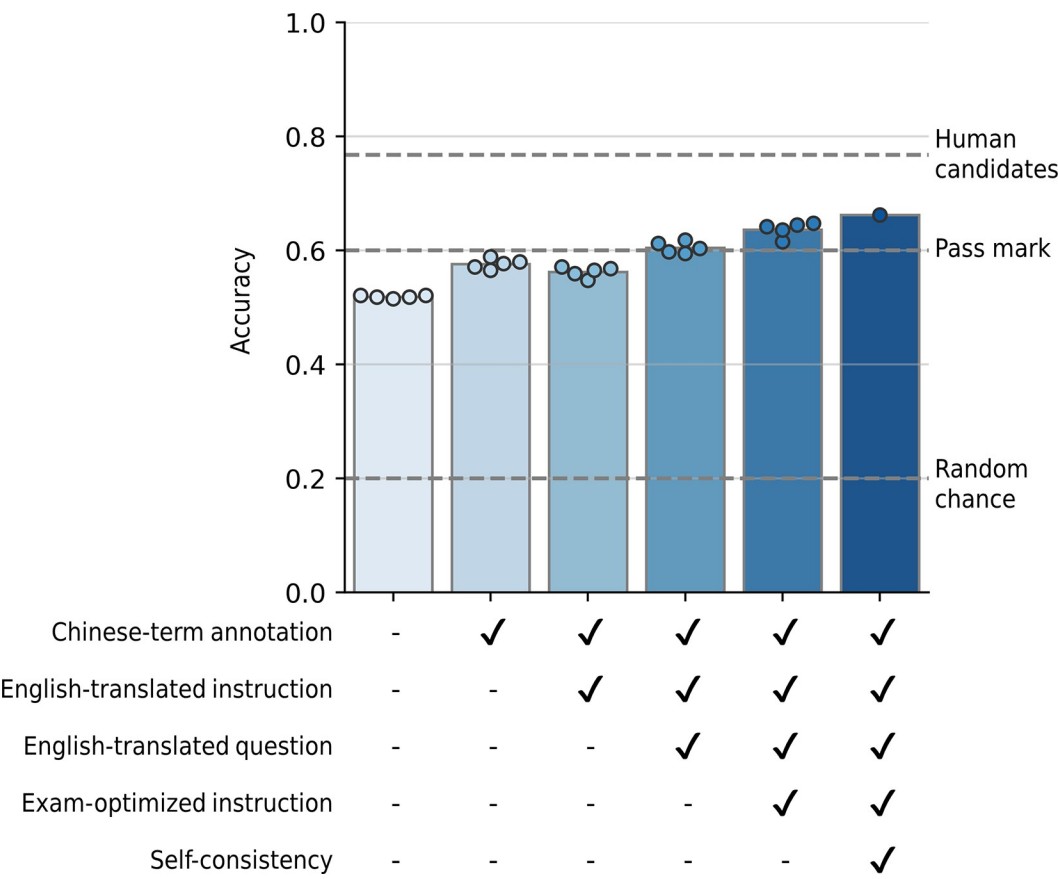

**Fig 1. The overall performance of GPT-4 with different prompt designs.** The x-axis and y-axis represent a prompt and the accuracy for the prompt, respectively. The heights of the bars represent the mean of accuracy for multiple trials. A circle mark indicates accuracy in each trial. Dashed lines indicate the chance level of accuracy of 20%, the pass mark of 60%, and the average accuracy of human candidates for the examination of 76.7%.

### 3.2. The difference in the performance across subjects

We analyzed the proficiency of GPT-4 across individual subjects. Although there was a variation in accuracy among subjects, the model consistently achieved an accuracy exceeding 40%, the pass mark for each subject, in all evaluated subjects. GPT-4 scored above the pass mark in 7 out of 12 subjects including herbology (accuracy of 87.5%), neuropsychiatry (81.2%), pediatrics (79.2%), gynecology (78.1%), internal medicine (1) (75.0%), physiology (75.0%), and preventive medicine (70.8%). On the other hand, accuracy on public health & medicine-related law (40.0%), internal medicine (2) (43.8%), and acupuncture medicine (52.1%) that are localized in Korea and TKM showed relatively low accuracy compared to other subjects, while the accuracies surpassed the pass mark of 40% for individual subjects (Fig 2). This suggests that GPT-4 would have lower performance on topics that are specialized in Korea and TKM.

### 3.3. TKM-specialized knowledge and inference

The significant disparity in performance across different subjects led us to hypothesize that the variations in accuracy by subject may be attributed to the extent of TKM-specialized knowledge and inference required for each. To validate this hypothesis, we measured the difference in accuracy between questions necessitating TKM-specialized knowledge and inference and

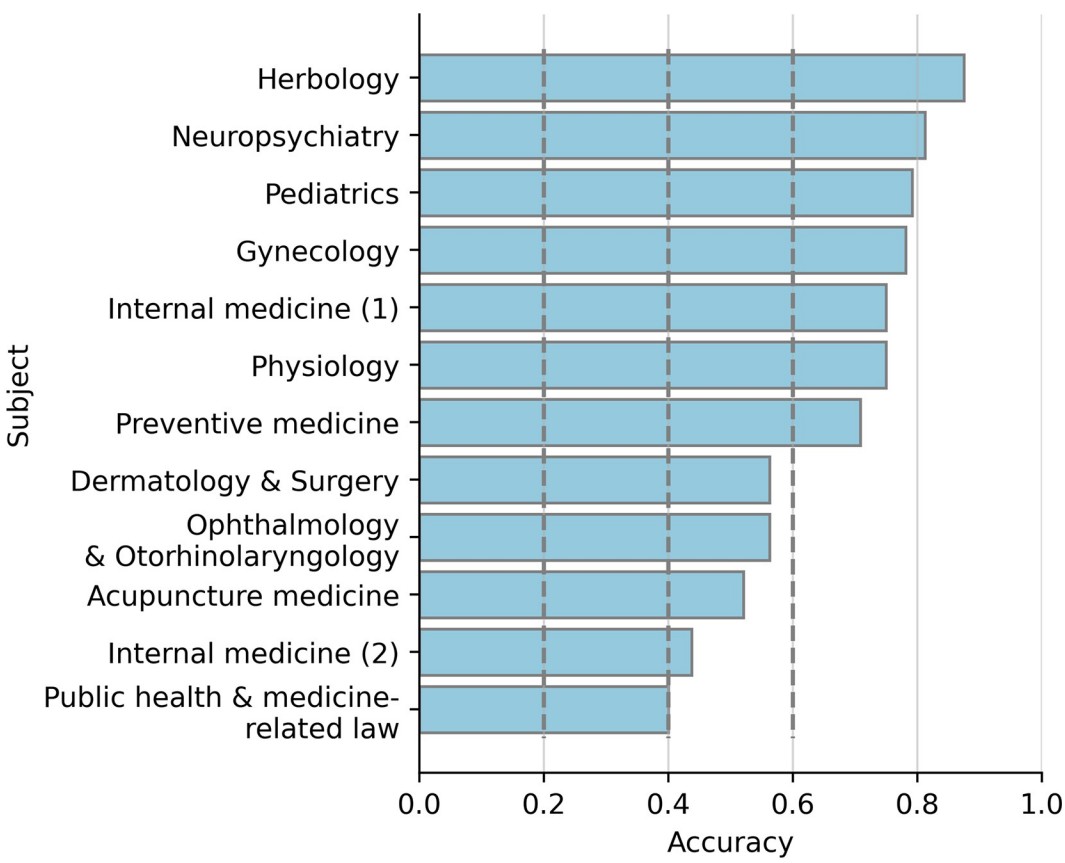

**Fig 2. The difference in accuracy between subjects.** The y-axis indicates the subjects the questions are related to, and the x-axis indicates the accuracy of answers to the questions. Dashed lines indicate the chance level of accuracy of 20%, the pass mark of 40% for individual subjects, and 60% for total questions. Other details are the same as in Fig 1.

those that did not. While questions not necessitating TKM knowledge achieved an accuracy of 82.4%, those requiring TKM expertise recorded a lower accuracy of 55.4% (Fig 2A). A subject-by-subject comparison revealed a consistent trend: accuracy was invariably diminished for questions demanding TKM knowledge compared to those that didn't (S2 Fig). This underscores the notion that GPT-4 may exhibit reduced efficacy in making inferences related to TKM.

### 3.4. Relationship between competency type and accuracy

To assess the GPT-4's strengths and weaknesses across different types of tasks, we examined the difference in accuracy by competency types the question was intended to assess. GPT-4 showed an accuracy of 85.9% for diagnosis-based questions, 63.2% for recall-based questions, and 53.5% for intervention-based questions Notably, the model achieved high accuracy for knowledge-based or diagnosis-based questions while showing low accuracy for intervention-based questions (Fig 3). However, we found a potential correlation between the model's lowered performance in intervention questions and the necessity for TKM-specialized knowledge in problem-solving (Table 2). Supporting this, the model exhibited an accuracy of 100% for intervention-based questions that didn't call for TKM expertise, as opposed to approximately accuracy of 50% for those that did (S3 Fig). These imply that the observed decline in GPT-4's accuracy on intervention-based questions might be more attributed to the necessity of TKM knowledge rather than inherent differences in question types.

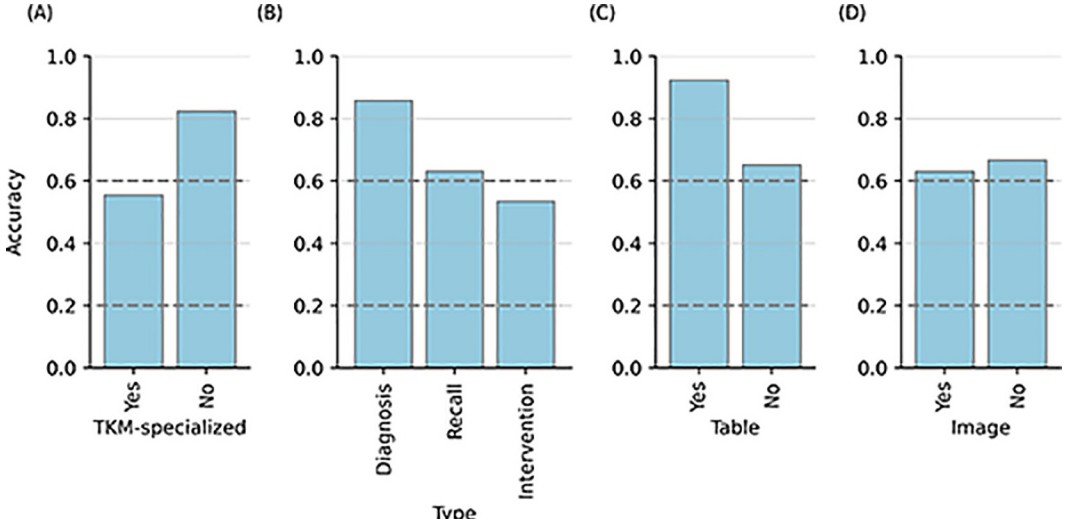

**Fig 3. The difference in accuracy according to the properties the questions have.** The x-axis indicates (A) whether TKM-specialized knowledge is required to answer the question, (B) what competency types the question is intended to assess, and (C) whether a question contains a table or (D) an image. The y-axis indicates the accuracy of the answers to the questions. Other details are the same as in Fig 1.

### 3.5. Effects of tables and images included in questions on the accuracy

Out of the 340 questions, 13 questions contain tables and 46 questions contain images, requiring the ability to interpret tables and images for answering correctly. However, since the models in this study only allow textual input, table inputs were limited to text separated by space, and image inputs were limited to the information that the question contained an image. To determine how these constraints affect performance, we analyzed the difference in accuracy between questions with tables or images and those without. While we initially anticipated potential limitations affecting its performance, GPT-4 achieved a higher accuracy of 92.3% for questions with tables, in comparison to 65.1% for those without, showing GPT-4's ability to process incompletely structured data. For questions that contained images, GPT-4 achieved an accuracy of 63.0%, exceeding the pass mark, even though in this study, GPT-4 couldn't actually 'view' the images. This suggests that the textual information provided to the model was adequate for predicting the correct response. Notably, this trend persisted irrespective of the necessity for TKM knowledge in the answers (S4 Fig, S5 Fig).

### 3.6. Consistency of response

In this study, we repeated five trials of the same question and obtained a response each trial. To investigate the consistency of responses of GPT-4 to the same questions, we analyzed how much the responses to the same question overlap each other in multiple trials. In this study, out of seven responses to the same question, the percentage of questions where the maximum number of identical responses ranged from two to seven was 0.29%, 5.29%, 13.82%, 14.71%, 15.88%, and 50.00%, respectively (Fig 4A). The accuracies for these corresponding categories were 28.57%, 23.02%, 32.52%, 43.43%, 52.65%, and 86.47%, indicating a positive correlation between response consistency and accuracy (Fig 4B). We suggest that the increase of accuracy by self-consistency in this study is associated with the high consistency of response for the correct prediction.

(A)                                        (B)

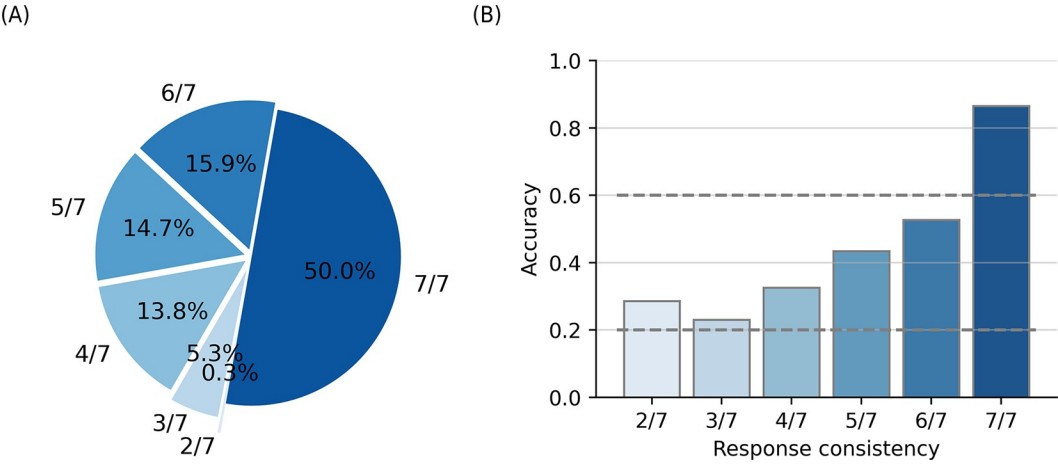

**Fig 4. The relationship between consistency of response and accuracy of the response in GPT-4.** (A) The percentage in the pie represents the percentage of questions according to corresponding response consistency. (B) The x-axis indicates the consistency of response, the number of responses with the same answers. The y-axis indicates the accuracy of the response for the questions with response consistency corresponding to the x-axis. Other details are the same as in Fig 1.

## 4. Discussion

Until recently, it was deemed essential to train models with additional massive biomedical-specific data to facilitate not only understanding biomedical texts but also making inferences using relevant knowledge [30–32]. However, well-trained general-purpose-based models have shown outstanding performance in natural language processing for biomedical applications with little or no fine-tuning on biomedical data. For example, GPT-3.5 and GPT-4 demonstrated accuracy over pass mark on the USMLE [23], ranked in the top one percent of scorers in the USA Biology Olympiad [22] and passed a surgery board examination [33] without fine-tuning. This suggests that the future development of biomedical AI for various specialties, including medicine, will deviate from the high-cost, low-efficiency approaches that dominated the past few decades. In this study, GPT-4 showed accuracy surpassing the pass mark of K-NLEKMD without the advanced training for TKM, suggesting that the application of the foundation models for TKM would be plausible.

In our study, however, it was also shown that the models exhibited weak performance on questions that require the ability to understand the Korean language and knowledge about TKM or Korea-adapted healthcare. This could be attributed to the limited depth of Korean language and TKM-related data in the training data of models. For example, the pre-training dataset for GPT-3, a model released in 2020 that GPT-3.5 is based on and the technical details have been officially reported, is mainly derived from a modified version of Common Crawl [19] in which English-based data accounts for about 50%, while Korean data accounts for only 0.65% [28]. Moreover, WebText2, the most heavily weighted of the five datasets used to train GPT-3.5 consists of user-written posts from Reddit [34]. Reddit has the largest number of users from the United States (47.13%), followed by predominantly English-speaking countries such as the United Kingdom, Canada, and Australia [35], implying the culturally biased output toward Western culture from LLMs observed in several studies [36,37] is due to the biased train dataset. This suggests that LLMs including GPT-3.5 and GPT-4 are likely to underrepresent minority cultures, especially Korean. Indeed, it was noted that the benchmark for medical LLMs is required to be expanded for multilingual evaluation [38]. In our study, we observed low accuracy when applying the model to public health and medicine-related law in Korea.

Similarly, in the Japanese Medical Licensing Examinations, it was noted that LLMs strongly recommended against certain medical practices in Japan, including euthanasia [39]. Both cases suggest that the model may have difficulties in adapting to localized healthcare and medical policies. This underscores the importance of fine-tuning the model using datasets that reflect culture-specific medicine. Furthermore, biases in LLMs, which emerge from patterns in their training data, could potentially lead to health disparities and harm, especially in socioeconomic or racial contexts [40,41]. This is of particular concern for East Asians who might be underrepresented in datasets primarily from North America and Europe and thus may be disproportionately affected by these biases. On the other hand, it's also worth noting that texts related to TKM or traditional Asian medicine predominantly originate from East Asian countries where Asians are the majority. This indicates the need for caution when using LLMs to make TKM inferences about non-Asian ethnicities. Such biases rooted in prior knowledge must be considered when applying transnationally developed LLMs to fields such as healthcare, including TKM.

We found that high consistency of response is associated with increased accuracy for questions. Indeed, LLMs are susceptible to hallucinations, where they describe untrue things as if they were true, and OpenAI employs reinforcement learning from human feedback [21] to mitigate this issue in GPT-3.5 [20]. It was also found that implementing self-consistency in the MultiMedQA improved multiple-choice performance, notably enhancing the Flan-PaLM 540B model's performance on MedQA and MedMCQA datasets by over 7% [38]. These results suggest that consistency of response might serve as an indicator of the model's reliability for its response.

LLMs offer potential in clinical support, medical education, and TKM research. In clinical contexts, doctors should be able to discuss patients' symptoms with LLMs and receive guidance on medical decisions [42]. In the realm of medical education, LLMs can make resources more accessible, enhance resource efficiency, and offer simulation-based learning experiences [43]. For TKM research, LLMs can aid in understanding decision processes and address data collection challenges in clinical informatics. However, there are several concerns. Further research is essential for TKM-specific insights and ensuring real-world effectiveness. There's also a pressing need to mitigate the risk of LLMs providing outdated or biased answers. In education, the potential for conveying biases or incomplete medical information remains a concern. When used for TKM research, their inherent biases might misrepresent or overly simplify traditional practices and concepts.

In this section, we outline the limitations of our study regarding the evaluation of LLM for TKM. It's important to note that K-NLEKMD, which we used for assessment, is designed to assess the skills of candidates across a diverse range of subjects. However, this examination, mainly targeting medical trainees, may not be appropriate for assessing competencies at the level of experienced clinicians. Also, the focus on closed-ended questions in this study offers a limited view of LLM's capability, without delving into the quality of its reasoning. Further study should consider applying expert evaluation to determine the suitability of LLMs for real-world clinical use. Although this study does not suggest that LLMs are ready for clinical use, it is significant in demonstrating the feasibility of applying LLMs in TKM with a certificated assessment of TKM knowledge and inference.

In conclusion, this study offers early findings on the application of LLM to TKM. We found that GPT-4 achieved performance that is enough to pass the K-NLEKMD without advanced training for TKM, suggesting the potential of the application of LLM to TKM. However, it was observed that language-dependent performance degradation and limitations in Korea-localized and TKM-specific areas, underscoring the need for careful consideration in the safe application of LLM to TKM and other culturally-adapted medicines. For the safe and effective use

of LLM in TKM, collaboration among healthcare providers, medical engineers, and policy-makers is essential.

## Supporting information

**S1 Fig. The overall performance of the GPT-3.5 and GPT-4 with Chinese-term annotation.** Note that the results shown in this figure are for the performance of the model when different prompting techniques including translating the question into English, providing exam-optimized instructions, and utilizing a self-consistency in the prompt were not used. The x-axis and y-axis represent the language model and its accuracy, respectively. The heights of the bars represent the mean of accuracy for multiple trials. The color represents the model (GPT-4 or GPT-3.5). A circle mark indicates accuracy in each trial. Dashed lines indicate the chance level of accuracy (= 0.2) and the pass mark (= 0.6).
(DOCX)

**S2 Fig. The difference in accuracy between subjects.** The y-axis indicates the subjects the questions are related to, and the x-axis indicates the accuracy on the questions. The colors of the bar indicate whether TKM-specialized knowledge is required to answer for the questions. Other details are the same as in Fig 2.
(DOCX)

**S3 Fig. The difference in accuracy between questions that require TKM knowledge and questions that do not, compared by competency types.** The x-axis indicates what competency types the question is intended to assess, and the y-axis indicates the accuracy on the questions. The colors of the bar indicate whether TKM-specialized knowledge is required to answer for the questions. Other details are the same as in Fig 1.
(DOCX)

**S4 Fig. The difference in accuracy between questions that require TKM knowledge and questions that do not, compared by whether the question includes tables or not.** Details are the same as in the S2 Fig.
(DOCX)

**S5 Fig. The difference in accuracy between questions that require TKM knowledge and questions that do not, compared by whether the question includes images or not.** Details are the same as in the S2 Fig.
(DOCX)

**S1 Table. Examples for prompt engineering in this study.**
(DOCX)

**S2 Table. An example of an original question and Chinese-annotated question input into GPT-4. Since the questions of the Korean national licensing examination for Korean medicine doctors used in the study are not publicly available, we show virtual questions as an example.** The original questions were all written in Korean, but we present both original questions written in Korean and translated questions in English for the reader's convenience. The underlines indicate Chinese-annotated TKM terms. TKM, traditional Korean medicine.
(DOCX)

## Author Contributions

**Conceptualization:** Dongyeop Jang, Chang-Eop Kim.

**Data curation:** Dongyeop Jang.

**Formal analysis:** Dongyeop Jang, Tae-Rim Yun, Chang-Eop Kim.

**Funding acquisition:** Young-Kyu Kwon, Chang-Eop Kim.

**Investigation:** Dongyeop Jang.

**Methodology:** Dongyeop Jang, Tae-Rim Yun, Chang-Eop Kim.

**Project administration:** Chang-Eop Kim.

**Software:** Dongyeop Jang, Tae-Rim Yun.

**Supervision:** Chang-Eop Kim.

**Visualization:** Dongyeop Jang.

**Writing – original draft:** Dongyeop Jang, Chang-Eop Kim.

**Writing – review & editing:** Choong-Yeol Lee, Young-Kyu Kwon, Chang-Eop Kim.

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
