## [Decision Letter · Decision Letter 0]

9 Aug 2023

PDIG-D-23-00147

Exploring the Potential of Large Language models in Traditional Korean Medicine: A Foundation Model Approach to Culturally-Adapted Healthcare

PLOS Digital Health

Dear Dr. Kim,

Thank you for submitting your manuscript to PLOS Digital Health. After careful consideration, we feel that it has merit but does not fully meet PLOS Digital Health's publication criteria as it currently stands. Therefore, we invite you to submit a revised version of the manuscript that addresses the points raised during the review process.

Please submit your revised manuscript within 60 days Oct 08 2023 11:59PM. If you will need more time than this to complete your revisions, please reply to this message or contact the journal office at digitalhealth@plos.org. Please include the following items when submitting your revised manuscript:

We look forward to receiving your revised manuscript.

Kind regards,

Luis Filipe Nakayama, M.D.

Guest Editor

PLOS Digital Health

Journal Requirements:

Additional Editor Comments (if provided):

Review for: Exploring the Potential of Large Language models in Traditional Korean Medicine: A Foundation Model Approach to Culturally-Adapted Healthcare.

This is an interesting manuscript that evaluates the performance of GPT3.5 and GPT 4 in the context of the Korean boarding exam, particularly focusing on their performance across traditional Korean medicine (TKM) questions.

Considerations:

- The manuscript's title suggests an exploration of the potential of LLMs in TKM, yet the article predominantly discusses LLM applications in the Korean licensure examination. I suggest either aligning the title with the article’s research question/methods/results/discussion and refining the manuscript content to better match the title.

- While the introduction section touches on TKM, the article’s methods and results do not primarily focus on the differences in TKM questions. Strengthening the introductory paragraphs to create a clearer link between TKM and the examination context would enhance coherence.

- Improvement is needed in the introduction and sections addressing sources of biases and limitations inherent to LLMs.

- If the questions are not publicly accessible, clarification is needed regarding the source of questions and answers. Additionally, information about any annulled questions or quality control measures would enhance transparency.

- Is there a comparative performance parameter between questions written in Chinese characters and those in Korean characters that supports the statement, “Considering that the language model may not be able to fully learn sentences containing TKM terms that are often written in Chinese characters but written only in Korean characters in the questionnaires”?

- If the manuscript aims to explore LLM capacity in TKM, an expansion of the description of TKM specialized knowledge questions and an analysis of their performance across different groups is suggested.

- Similarly, Table 2 could be divided to categorize TKM-specialized knowledge questions as yes/no and include the number of questions related to competency/table/images. Additionally, an analysis across TKM/non-TKM specialized knowledge questions (Figure 3) would enhance clarity.

- Regarding repeated questions, was a consistent answer required across all five repetitions for it to be considered final? Clarification is needed, especially considering the potential impact of an incorrect answer compared to a correct one. e.g., if the answer was 4 times correctly answered and 1 time wrong, was the answer considered wrong?

- I recommend including the number and details of questions encompassing TKM-specialized knowledge in the text, not solely within the table.

- The discussion about the lower performance of LLMs in TKM questions and the reasons discussed in the discussion section, such as limited depth in algorithm training data and insufficient scientific articles on TKM, should be expanded upon. Do the authors believe that applying LLM-based models as clinical assistants, for medical education, or TKM research would benefit or harm the Korean population?

- It is common for medical exams to present straightforward and typical clinical scenarios that can potentially inflate LLM performance in healthcare. Does this phenomenon occur in the Korean licensure examination? Moreover, is LLM a reliable source of TKM information for Korean patients?

- I suggest adding to the discussion section about the risks associated with deploying LLM-reliant systems for Korean healthcare/TKM. Considering that LLMs are trained on less than 1% of Korean data, they pose a potential risk to the Korean population.

Reviewers' comments:

Reviewer's Responses to Questions

**Comments to the Author**

1. Does this manuscript meet PLOS Digital Health’s publication criteria? Is the manuscript technically sound, and do the data support the conclusions? The manuscript must describe methodologically and ethically rigorous research with conclusions that are appropriately drawn based on the data presented.

Reviewer #1: Yes

Reviewer #2: Yes

Reviewer #3: Yes

2. Has the statistical analysis been performed appropriately and rigorously?

Reviewer #1: Yes

Reviewer #2: Yes

Reviewer #3: Yes

3. Have the authors made all data underlying the findings in their manuscript fully available (please refer to the Data Availability Statement at the start of the manuscript PDF file)?

Reviewer #1: Yes

Reviewer #2: No

Reviewer #3: No

4. Is the manuscript presented in an intelligible fashion and written in standard English?

Reviewer #1: Yes

Reviewer #2: Yes

Reviewer #3: Yes

5. Review Comments to the Author

Reviewer #1: Thank you for submitting such an interesting paper. I would like to encourage you to consider the following improvements to your paper for publication. 

Throughout the paper, the authors directly embedded Korean questions into GPT 3.5 and GPT4.0 to solve the test questions. 

As a result, the hypotheses of this study becomes a mixture of two topics. 

First, how accurately do GPT 3.5 and GPT4.0 translate TKM-related questions in Korean or Korean+Chinese?

Second, how well do GPT 3.5 and GPT4.0 understand and solve TKM-related questions (assuming accurate translation)?

The two issues must be distinguished, otherwise, the purpose of this study will be obscured. 

In the Discussion, the authors extrapolate the problem of language as the reason why LLMs scored lower on the K-NLEKMD's test. 

This issue could be clarified by having GPT 3.5 and GPT4.0 translate the K- NLEKMD and then having TKM experts evaluate the quality of the translation. 

If 5 to 10 representative questions per subject and per feature were selected for translation and evaluation, the authors would be able to provide accurate evidence rather than vague guesses. 

Furthermore, if the same study were conducted with a sufficiently well-translated test paper in English, it would be possible to evaluate how well GPT 3.5 and GPT 4.0 are able to address TKM knowledge without prior training. 

This is an example of how the two hypotheses that are currently conflated can be interpreted separately. Please refer to the above methods and present your hypotheses, methods, and results more clearly. 

If you have difficulty translating the full text of the K-NLEKMD into English, I recommend that you make it clear in the first question; how much the translation problem may have affected the accuracy of the question. 

For the main body of the paper, please consider the comments below for revision. 

In section 3.1, the meaning of 'questions (native) without the labeling' is unclear, and it is unclear what 'former case' means. Please describe them more clearly. 

In section 3.4, it is not clear how you distinguish between 'TKM specialized knowledge'. Please add what process you used to distinguish whether 'TKM specialized knowledge' was included in the question or not. 

It is also recommended to analyze the questions at least in three categories: 1) questions that ask for modern medical knowledge (for which sufficient performance is already known), 2) questions that ask for traditional Chinese medicine knowledge (for which this study actually wants to know), and 3) questions that mix traditional Chinese medicine and modern medical knowledge.

Reviewer #2: Introduction: 

The introduction effectively introduces the background of Traditional Korean Medicine and the challenges of applying AI modeling due to limited data and implicit processes. It highlights the potential of large language models (LLMs) like GPT-3.5 and GPT-4, which have shown impressive performance in medical knowledge despite not having medicine-specific training.

Methodology: 

The methodology section describes the dataset used for evaluation, the specific subjects and questions included in the test, the LLMs utilized (GPT-3.5 and GPT-4), and the evaluation process. 

In section 2.1. K-NLEKMD: Since the prompts used for model training are not publicly available, authors must be more detailed about what the dataset is made up of. Details should be added such as how many questions there are for each existing competence or for each subject or if the different subdivisions have already been defined by the researchers or are already part of the exam.

Results: 

The results section presents the performance of GPT-3.5 and GPT-4 on the medical test. It reports accuracy scores for both models, highlighting that GPT-4 achieved near-passing performance. The analysis of performance across subjects, competency types, and the impact of tables and images on accuracy provides valuable insights. However the paper presents the accuracy results of GPT-3.5 and GPT-4 without comparing them to any baseline, values such as the average performance of test applicants or existing AI approaches in the field of TKM (if they exist).

Discussion: 

The discussion interprets the results and offers potential explanations for variations in accuracy. It acknowledges the limitations of the models, particularly in handling TKM-specific knowledge, and points to the need for culturally-adapted training data to improve performance.

Cultural Bias: The paper mentions that GPT-4 showed limitations that were believed to be caused by culturally-biased learning. However, the specific examples or instances of cultural bias in the model's responses are not elaborated upon. Providing concrete examples of such biases would strengthen the paper's argument and highlight the need for culturally-adapted training data.

The paper uses questions from the 2022 K-NLEKMD for the evaluation of GPT-3.5 and GPT-4. However, there is no discussion about the representativeness of this dataset in terms of the diversity and complexity of Traditional Korean Medicine. The authors should justify why this particular dataset was chosen and discuss potential biases or limitations associated with it.

Conclusion:

The performance of the model do not seem to be entirely encouraging in some of the tests, demonstrating how in some cases the use of models such as GPT-3.5 or GPT-4 can also be harmful for the health systems of underrepresented groups. Both in the discussion and in the conclusions, the risks of these biases must be mentioned.

The conclusion should include a more comprehensive discussion of future directions for research. This could involve addressing the identified limitations, exploring ways to mitigate cultural biases, and proposing potential collaborations with experts in Traditional Korean Medicine for fine-tuning and enhancing the model's performance.

Reviewer #3: In this paper, the authors evaluate the utility of GPT-3.5 and GPT-4 in answering questions for Traditional Korean Medicine with the Korean National Licensing Examination for Korean Medicine Doctors.

The authors evaluate the overall performance, and perform an error analysis of different subjects, consistency, etc.

Strengths:

1) Paper is well-written and easy to read

2) Authors performs analyses from different viewpoints (e.g., sensitivity, consistency, subgroup performance)

3) Authors consider how answers are scored: e.g., in borderline cases where option may not be correctly mentioned, questions contain text/images

4) Authors consider if the questions are present in the training data

Weaknesses:

1) The protocol for marking answers that do not answer with the option number correctly is confusing: how is it judged to be "similar enough" to the correct answer?

2) It is not clear how images are passed to the model -- perhaps an illustrative image/overall flowchart would help?

3) The rationale for the specific prompt chosen (i.e., "..for research purposes...") is not clear. Have the authors experimented with additional prompts?

6. PLOS authors have the option to publish the peer review history of their article (what does this mean?). If published, this will include your full peer review and any attached files.

**Do you want your identity to be public for this peer review?** For information about this choice, including consent withdrawal, please see our Privacy Policy.

Reviewer #1: No

Reviewer #2: Yes: David Restrepo

Reviewer #3: No

---

## [Decision Letter · Decision Letter 1]

20 Nov 2023

GPT-4 can pass the Korean National Licensing Examination for Korean Medicine Doctors

PDIG-D-23-00147R1

Dear Kim,

We are pleased to inform you that your manuscript 'GPT-4 can pass the Korean National Licensing Examination for Korean Medicine Doctors' has been provisionally accepted for publication in PLOS Digital Health.

Best regards,

Luis Filipe Nakayama, M.D.

Guest Editor

PLOS Digital Health

All my comments have been successfully addressed.

Reviewer Comments (if any, and for reference):

Reviewer's Responses to Questions

**Comments to the Author**

1. If the authors have adequately addressed your comments raised in a previous round of review and you feel that this manuscript is now acceptable for publication, you may indicate that here to bypass the “Comments to the Author” section, enter your conflict of interest statement in the “Confidential to Editor” section, and submit your "Accept" recommendation.

Reviewer #1: All comments have been addressed

Reviewer #2: All comments have been addressed

2. Does this manuscript meet PLOS Digital Health’s publication criteria? Is the manuscript technically sound, and do the data support the conclusions? The manuscript must describe methodologically and ethically rigorous research with conclusions that are appropriately drawn based on the data presented.

Reviewer #1: Yes

Reviewer #2: Yes

3. Has the statistical analysis been performed appropriately and rigorously?

Reviewer #1: Yes

Reviewer #2: Yes

4. Have the authors made all data underlying the findings in their manuscript fully available (please refer to the Data Availability Statement at the start of the manuscript PDF file)?

Reviewer #1: No

Reviewer #2: No

5. Is the manuscript presented in an intelligible fashion and written in standard English?

Reviewer #1: Yes

Reviewer #2: Yes

6. Review Comments to the Author

Reviewer #1: The revised menuscript is well organized enough for publication.

Reviewer #2: The authors have effectively addressed all my comments and have justified the limitations while trying to answer some of my questions effectively, significantly improving the overall quality and robustness of the paper.

7. PLOS authors have the option to publish the peer review history of their article (what does this mean?). If published, this will include your full peer review and any attached files.

**Do you want your identity to be public for this peer review?** For information about this choice, including consent withdrawal, please see our Privacy Policy.

Reviewer #1: No

Reviewer #2: **Yes: **
